# Observation of nonlinear response and Onsager regression in a photon Bose-Einstein condensate

Alexander Sazhin [1], Vladimir N. Gladilin [2], Andris Erglis [3], Göran Hellmann [1,4], Frank Vewinger [1], Martin Weitz [1], Michiel Wouters [2] & Julian Schmitt [1] ✉

The quantum regression theorem states that the correlations of a system at two different times are governed by the same equations of motion as the single-time averages. This provides a powerful framework for the investigation of the intrinsic microscopic behaviour of physical systems by studying their macroscopic response to a controlled external perturbation. Here we experimentally demonstrate that the two-time particle number correlations in a photon Bose-Einstein condensate inside a dye-filled microcavity exhibit the same dynamics as the response of the condensate to a sudden perturbation of the dye molecule bath. This confirms the regression theorem for a quantum gas, and, moreover, demonstrates it in an unconventional form where the perturbation acts on the bath and only the condensate response is monitored. For strong perturbations, we observe nonlinear relaxation dynamics which our microscopic theory relates to the equilibrium fluctuations, thereby extending the regression theorem beyond the regime of linear response.

The application of linear response theory to systems that are subject to perturbations lies at the heart of many fundamental phenomena in physics[1], including electromagnetic wave propagation in optical media, structure factors in condensed matter systems, or superfluid phases in quantum gases[2–4]. For strong perturbations, the extension of this concept to nonlinear response has facilitated our understanding of ubiquitous effects such as higher-harmonic generation in optics[5] or the emergence of turbulent flow in cold-atom and exciton-polariton systems driven far from equilibrium[6,7]. When considering equilibrium systems, the linear response behaviour is commonly governed by the fluctuation-dissipation theorem[8], which states that the intrinsic fluctuations of a system are connected to the absorptive part of a response function by thermal energy. The Onsager-Lax theorem, covering situations where the magnitude of the fluctuations is small, remains valid also for systems out of equilibrium and describes a universal relationship between the correlations and the response dynamics, as has been theoretically shown for irreversible processes in classical and quantum systems[9–17]. The usual regression theorem, which states that the two-time averages $\langle A(t)B(0)\rangle$ of two observables $A$ and $B$ obey the same equations as the one-time averages $\langle A(t)\rangle$ for Markovian systems[12], links the system's fluctuations to the linear response, but more recent works have theoretically addressed nonlinear response as well[18]. Experimentally, it has been indirectly verified by measurements of Onsager's reciprocity relations in the classical domain, e.g., in thermoelectric systems or thin films[19,20]. A direct experimental test of this central theorem of statistical physics, however, by independent measurements of the temporal response to a perturbation and of the system's fluctuation dynamics has so far not been carried out for quantum gas systems[21–23].

To examine the regression theorem for a quantum gas, we investigate the reservoir-induced dynamics of a Bose-Einstein condensate (BEC) of photons in a dye-filled optical microcavity. In the used experimental platform, a two-dimensional photon gas is coupled radiatively to a reservoir of dye molecules[24–26]. Previous work has, by

[1]Institut für Angewandte Physik, Universität Bonn, Bonn, Germany. [2]TQC, Universiteit Antwerpen, Antwerpen, Belgium. [3]Physikalisches Institut, Albert-Ludwigs-Universität Freiburg, Freiburg, Germany. [4]Present address: Leibniz Institute of Photonic Technology, Jena, Germany. ✉e-mail: schmitt@iap.uni-bonn.de

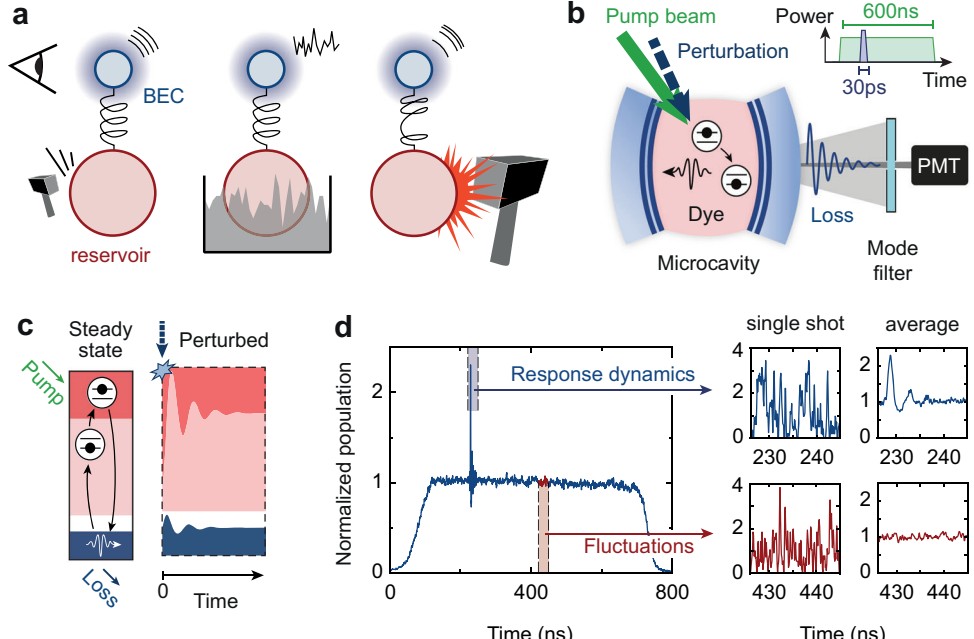

**Fig. 1 | Experimental scheme to probe the regression theorem. a** Mechanical analogue of a photon Bose-Einstein condensate (top, blue) coupled to a molecule reservoir (bottom, red), forming a quantum dissipative oscillator. A weak sudden perturbation of the reservoir (left) leads to a linear response of the average single-time photon number $\langle n(t) \rangle$. Immersion into a "coloured-noise" bath (middle) leads to number fluctuations $\delta n$, or two-time correlations $\langle \delta n(t) \delta n(0) \rangle$, for which the regression theorem predicts the same dynamics. For a strong perturbation (right), the response becomes nonlinear. **b** Dye-filled optical microcavity with photon Bose-Einstein condensate, which is perturbed by a laser pulse irradiated on the dye reservoir (inset). The fluctuation and response dynamics are recorded on a photomultiplier (PMT). **c** In a steady state, when cavity losses are compensated by pumping the dye, photons (blue) and molecules in the ground and excited state (red) are in equilibrium due to absorption and emission events. After a sudden perturbation of the excited molecules, also the condensate population is perturbed and both systems relax back to their stationary values. **d** Left: Temporal evolution of the condensate averaged over several time traces. Right: Zoom in on the time window used to analyse the photon number response dynamics (top row), and the time window used for the fluctuation analysis (bottom). The left panels show single-shot data and the right panels the averaged data.

observing the statistical number fluctuations that result from an effective particle exchange with the molecular reservoir, reported a non-Hermitian phase transition[27,28], verified a fluctuation-dissipation relation connected to a reactive response function[29], and realised protocols to temporally perturb the reservoir[30,31]. Moreover, theory work on this system has employed the regression theorem to calculate two-point correlations from the relaxation dynamics[32–34].

Figure 1 illustrates how a photon BEC couples to a reservoir of dye molecules, providing a benchmarking platform for the regression theorem in complex many-body quantum systems of both matter and light. Uniquely, the system allows one to connect the linear response dynamics after a weak perturbation (Fig. 1a, left panel) with the intrinsic fluctuations driven by coloured noise (Fig. 1a, middle) and with the nonlinear response after a strong perturbation (Fig. 1a, right). Unlike the more usually encountered situation for the quantum regression theorem where a system (here the BEC) itself is perturbed, the perturbation here acts on the reservoir part (molecules), the excitation is transferred to the system by light-matter interactions (illustrated by a spring), and only then the photon number dynamics is witnessed by an observer before it is damped out by dissipation of photons to the environment. Despite its microscopic complexity, at the mean-field level, the photon-molecule system can effectively be described as a damped harmonic oscillator, which exhibits a pronounced nonlinearity for large perturbations.

In this study, we measure the nonequilibrium response dynamics of a photon Bose-Einstein condensate after a sudden perturbation of its equilibrium dye reservoir. By comparing the response to the independently measured number fluctuations of the condensate, we first experimentally confirm the validity of the regression theorem for the optical quantum gas in the limit of weak perturbations. Further, for stronger perturbations, we observe the emergence of nonlinear dynamics that are attributed to the saturation of the particle reservoir. Notably, the seemingly violated regression theorem is restored by a theoretical model that captures the nonlinear response dynamics, whose relevant parameters are the same as in the linear response. Such a nonlinearity in the condensate-bath system, which has been theoretically predicted also for exciton-polariton systems[35], forms the basis for future studies into the properties of elementary and topological excitations within lattices of photon condensates[36–38].

## Results

### Experimental scheme

Our photon Bose-Einstein condensates are prepared inside an optical microcavity filled with a dye solution of refractive index $\tilde{n} \approx 1.44$ and concentration 1 mmol L$^{-1}$, see Fig. 1b; for details see ref. 32 and Methods. The microcavity is formed by two spherical mirrors with a reflectivity > 99.998% and a 1 m radius of curvature. At the used microcavity length $D_0 \approx 1.5\,\mu m$, the free spectral range of the cavity becomes as large as the emission and absorption spectral profiles of the dye medium, which restricts the photon dynamics to the two transverse degrees of freedom at a fixed longitudinal mode number $q = 7$. Inside the microcavity, the photons behave as a two-dimensional gas of bosons with effective mass $m_{ph} = \pi \hbar q \tilde{n}/(D_0 c) \approx 10^{-35}$ kg and quadratic dispersion; the minimum photon energy $m_{ph}(c/\tilde{n})^2 = \hbar \omega_c \approx 2.1$ eV is given by the energy of the transverse ground mode. In addition, the mirror curvature induces a harmonic trapping potential of frequency $\Omega/(2\pi) \approx 40$ GHz for the photons. The emission and absorption rates $B_{em}(\omega)$ and $B_{abs}(\omega)$ of the dye medium at $T = 300$ K fulfil the Kennard-Stepanov relation $B_{em}/B_{abs} \propto \exp(-\hbar \Delta / k_B T)$, which depends on the detuning $\Delta = \omega - \omega_{zpl}$ of the photon frequency from the zero-phonon line $\omega_{zpl}$[27]. By

absorption-emission cycles with dye molecules ($10^{-12}$ s timescale), the photons thermalise to the rovibronic temperature of the dye before leaving the cavity ($10^{-9}$ s). Above the critical photon number $N_c = \pi^2/3(k_B T/\hbar\Omega)^2 \approx 80000$, the photon gas exhibits Bose-Einstein condensation[24–26]. Despite the thermalisation mechanism, the photon Bose-Einstein condensate realises a weakly dissipative macroscopic quantum system due to losses, e.g., from photons leaking through the cavity mirrors. To establish a steady state of photons and molecules, the dye medium is externally pumped.

## Theoretical description

Under steady-state conditions, the weakly dissipative character of the open condensate system is evident from an exceptional point in the second-order temporal correlations[28,34]. The underlying stochastic number fluctuations are caused by an effective particle exchange between the condensate and a large reservoir of excited molecules[27]. In the present work, we access the open-system dynamics of the condensate by investigating the photon number response after a controlled sudden perturbation of this reservoir. This allows us to test the validity of the regression theorem for the optical quantum gas, as well as to explore it beyond the regime of linear response. To begin with, we derive an analytical expression for the response dynamics[39] from two coupled rate equations for the number of photons $n$ and excitations $X = n + M_\uparrow$, respectively, given by $dn/dt = B_{em}M_\uparrow(1 + n) - B_{abs}M_\downarrow n - \kappa n$ and $dX/dt = PM_\downarrow - \kappa n - \Gamma_{sp}M_\uparrow$ (see Methods). Here $M_{\downarrow,\uparrow}$ denotes the number of molecules in their ground ($\downarrow$) and electronically excited ($\uparrow$) states, $P$ the pump rate, $\kappa$ the cavity loss rate, and $\Gamma_{sp}$ the spontaneous decay rate to unconfined modes. A steady state is established if the losses are compensated for by a constant pumping of rate $P = \kappa\langle n\rangle/\langle M_\downarrow\rangle + \Gamma_{sp}\langle M_\uparrow\rangle/\langle M_\downarrow\rangle$, where $\langle\ldots\rangle$ denotes temporal averaging, see Fig. 1c.

A time-dependent perturbation $P(t)$ of the molecule reservoir drives the photon condensate away from the steady state. The resulting photon number evolution $n(t) = \langle n\rangle \exp\{B_{em}[1 + \exp(\hbar\Delta/k_B T)]\int_0^t m(t')dt'\}$ depends on the strength of the perturbation and therefore on the deviation $m(t)$ of the excited molecule number from the steady state number that would correspond to the instantaneous photon number $n(t)$ (see Methods). The implicit interplay between $m(t)$ and $n(t)$ also implies that a subsequent variation of the photon number leads to a change in the number of excited molecules. As detailed in the Methods, approximating the time integral over the perturbed molecules yields a nonlinear expression for the photon number response $R(t) = n(t) - \langle n\rangle$:

$$R(t) = \langle n\rangle \exp\left[B_{em}\left(1 + e^{\frac{\hbar\Delta}{k_B T}}\right)m_0 \frac{e^{s_+ t} - e^{s_- t}}{s_+ - s_-}\right] - \langle n\rangle \quad (1)$$

Here $m_0 = m(0)$ gives the number of molecules excited by the pulse perturbation at $t = 0$. The quantities $s_\pm = -\delta \pm \sqrt{\delta^2 - \omega_0^2}$ depend on the system parameters $\Delta, B_{em}, T, \Gamma_{sp}, \kappa$, molecule number $M$, and the steady-state photon number $\langle n\rangle$, where the damping rate $\delta$ and the oscillation frequency $\omega_0$ determine whether a biexponentially damped ($\delta > \omega_0$) or an oscillatory ($\delta < \omega_0$) response is expected. Note that $s_\pm$ are the eigenvalues of the non-Hermitian matrix which describes the linearised equations of motion for the variation of the photon number $n(t) - \langle n\rangle$ and excitation number $X(t) - \langle X\rangle$; for a discussion see the Methods and ref. 28. Expanding eq. (1) yields a linear expression for the response $R(t) \simeq \langle n\rangle B_{em}[1 + \exp(\hbar\Delta/k_B T)]m_0(e^{s_+ t} - e^{s_- t})/(s_+ - s_-)$. Both expressions are used to analyse the response dynamics, but eq. (1) describes the measured response more accurately in the regime of strong perturbations, while the linearisation is valid in the limit of small $m_0$. The expected second-order coherence at time delay $\tau$ on the other hand can, by utilising the regression theorem, be written as $g^{(2)}(\tau) \propto (s_+ e^{s_+ \tau} - s_- e^{s_- \tau})/(s_+ - s_-) \propto dR(\tau)/d\tau$ (see Methods). Physically, the

derivative follows from the fact that it takes some time for the molecules excited by the laser pulse to be converted into photons. This gives a delay for the response function concerning the correlation function, in contrast to a direct perturbation of the photon number. Unlike the response, the fluctuation dynamics $g^{(2)}(\tau)$ is intrinsically linear due to the grand canonical condition for large fluctuations being realised only for large reservoirs which basically cannot be saturated.

## Experimental protocol

To probe the response and fluctuation dynamics, we first prepare a steady-state photon BEC by quasi-cw optical pumping of the dye molecule reservoir over 600 ns at 532 nm wavelength, see Fig. 1b, c. After roughly 200 ns, a short laser pulse of 28 ps duration (also at 532 nm) irradiates the microcavity to perturb the reservoir. Part of the emission leaking from the microcavity is filtered in momentum space and for polarisation, and the photon number evolution in the transmitted condensate mode is recorded using a photomultiplier (see Methods). Figure 1d shows an example of condensate evolution after averaging over many time traces, from which the response dynamics are visible in a time window of 20 ns after the perturbation. The fluctuation dynamics are determined by analysing the second-order correlations $g^{(2)}(\tau)$ from individual time traces. Throughout all measurements the system parameters $B_{em} = 25$ kHz, $\hbar\Delta = -3.87 k_B T$, and $\Gamma_{sp} = 200$ MHz remain fixed, while the total molecule number $M = 5.4(15)\cdot 10^9$, and mirror transmission $\kappa = 6.4(10)$ GHz are obtained from fits to the data.

## Second-order correlation and response dynamics

Figure 2 shows measured second-order correlation functions $g^{(2)}(\tau)$ and photon number responses $R(t)$ along with fits for two steady-state photon numbers $\langle n\rangle$, realised by varying the quasi-cw pump power. We are first interested in the linear response of the photon condensate, so we use only relatively small perturbation pulse powers that weakly change the condensate population by $\delta n(t_{max})/\langle n\rangle = 0.31(7)$ on average, where $t_{max}$ denotes the time when the photon population has reached its maximum. Accordingly, the response data is fitted using the linearised form of $R(t)$ discussed above. While generically $g^{(2)}(\tau) = 1$ and $R(t) = 0$ are found for large $\tau$ and $t$, both data sets display distinct dynamics. For small $\langle n\rangle = 1740$, both $g^{(2)}(\tau)$ and $R(t)$ decay biexponentially, while for large $\langle n\rangle = 14250$ a damped oscillation of the fluctuations and the response is observed. The quantitative agreement of the dynamics for each photon number is seen in the respective values of the parameters $\delta, \omega_0$, see Fig. 2c, which determine the eigenvalues $s_\pm$. This gives evidence that the intrinsic number fluctuations and the response dynamics of the photon condensate to an external perturbation of the reservoir are governed by the same microscopic physics. Qualitatively, the agreement is visible when forming the derivative of the fitted response $dR(t)/dt$, see the inset of Fig. 2b, which well resembles the corresponding fit for $g^{(2)}(\tau)$ from Fig. 2a. Note that the biexponential and oscillatory dynamics are distinctly characterised by real and complex-valued $s_\pm$, respectively, which allows to identify a transition point between both dynamical regimes at the degeneracy $s_+ = s_-$, known as an exceptional point[28].

## Regression theorem in linear regime for weak perturbations

To systematically verify the universal relationship between $R(t)$ and $g^{(2)}(\tau)$ in the linear response regime, we next study the eigenvalues $s_\pm$ of the condensate dynamics as a function of the steady-state population $\langle n\rangle$. Figure 3a shows the measured damping rate $\mathrm{Re}(s_\pm)$ and oscillation frequency $\mathrm{Im}(s_\pm)/(2\pi)$ for the response and fluctuation dynamics (symbols), respectively, along with the eigenvalue prediction from eq. (1) (lines). Note for the response $R(t)$ we only show data for the smallest experimentally realised perturbation powers (as in Fig. 2), where the linear model is expected to be well applicable. At photon numbers below the one at the exceptional point $\langle n\rangle_{EP} \approx 2000$, two

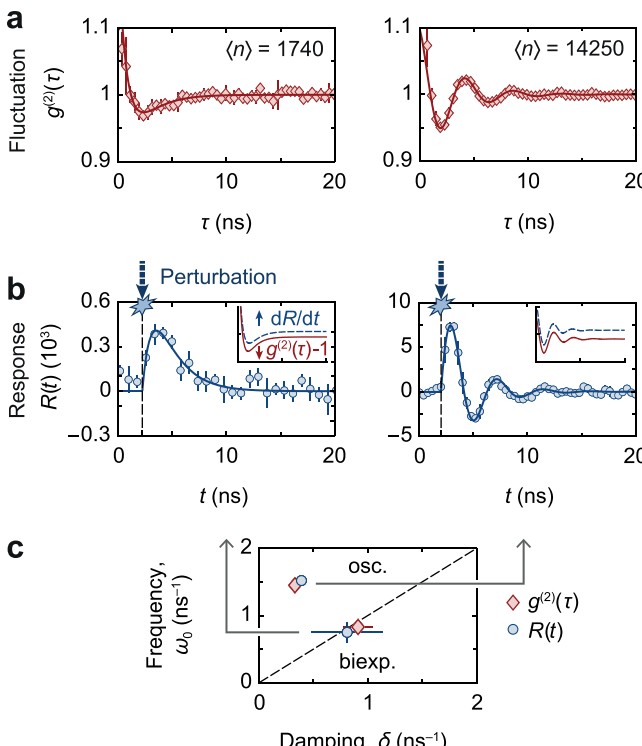

**a**

**b** Perturbation

**c**

**Fig. 2 | Fluctuation and response dynamics. a** Second-order correlation functions $g^{(2)}(\tau)$ for steady-state photon numbers $\langle n \rangle = 1740$ (left) and 14250 (right). Solid lines show the fits based on which the biexponential or oscillatory dynamics are identified; for details on fitting see Methods. **b** Temporal evolution of the photon number response in the condensate after a weak pulsed perturbation of the reservoir at the same $\langle n \rangle$ as in (**a**). The arrival time of the perturbation pulse is indicated by dashed vertical lines. Similarly, the dynamics are biexponential or oscillatory as determined by fitting the linear expression for $R(t)$ shown as solid lines. Insets show the scaled derivative of the fitted response $dR/dt$ and fitted $g^{(2)}(\tau) - 1$, vertically shifted for clarity. **c** Fit parameters in $\delta$-$\omega_0$ plane from (**a**) $\delta_{\text{fluct}} = \{0.91(13), 0.33(3)\}$ ns$^{-1}$ and $\omega_{0,\text{fluct}} = \{0.83(6), 1.45(2)\}$ ns$^{-1}$ (red diamonds), and from (**b**) $\delta_{\text{resp}} = \{0.81(32), 0.39(2)\}$ ns$^{-1}$ and $\omega_{0,\text{resp}} = \{0.75(14), 1.52(2)\}$ ns$^{-1}$ (blue circles). The dashed line separates the parameter regions with biexponential ($\delta > \omega_0$) or oscillatory ($\delta < \omega_0$) relaxation dynamics, respectively. Error bars show standard statistical errors in (**a**) and (**b**), and standard fitting errors in (**c**).

branches of damping constants with vanishing oscillation frequency indicate the biexponential regime, while for larger photon numbers the observed merging of $\text{Re}(s_\pm)$ and bifurcation in $\text{Im}(s_\pm)$ highlight the regime of oscillatory dynamics.

Figure 3 can be understood as a nonequilibrium phase diagram of the photon dynamics, where $\langle n \rangle$ presents a control parameter to tune between both phases. The associated opening of a gap in the complex plane along the imaginary axis at the phase transition is visible in the complex eigenvalue spectra of $g^{(2)}(\tau)$ and $R(t)$ in Fig. 3b; symbol colours indicate the photon number, which parametrises the spectral trajectory of $s_\pm(\langle n \rangle)$. As the photon number is increased, the real-valued $s_\pm$ (biexponential dynamics) move from $\{\text{Re}(s_+),\text{Im}(s_+)\} = \{-\infty,0\}$ns$^{-1}$ and $\{\text{Re}(s_-),\text{Im}(s_-)\} = \{0,0\}$ns$^{-1}$ along the real axis and coalesce at the exceptional point $\{\text{Re}(s_\pm),\text{Im}(s_\pm)\} \approx \{-0.8,0\}$ns$^{-1}$. As $\langle n \rangle$ is increased further, the eigenvalues $s_\pm$ separate again and move into the imaginary plane (oscillatory dynamics).

The agreement between the measured linear response, the fluctuation dynamics, and the theory prediction provides an experimental benchmark for the regression theorem in optical quantum gases. Direct evidence for this conclusion is given in Fig. 3c, which compares the eigenvalues $s_\pm$ of the response measurement to the values of $s_\pm$ obtained from the fluctuation measurement for pairwise corresponding photon numbers. In order to represent the two sets ($s_+$ and $s_-$) of

complex-valued data, we show the four contributions $\text{Re}(s_+)$, $\text{Re}(s_-)$, $\text{Im}(s_+)$ and $\text{Im}(s_-)$. We find excellent agreement between both sets of eigenvalues, as evident from the data aligning with a linear curve of slope one (black line). Interestingly, for large condensate populations currently not accessible in the experiment and shown in Fig. 3d, our theoretical model for the photon dynamics predicts a second exceptional point where again $s_+ = s_-$, or equivalently $\delta = \omega_0$ (see Methods for the corresponding functions). Physically, the re-emerging biexponential phase results from the different asymptotic scalings of $\delta \sim \langle n \rangle$ and $\omega_0 \sim \sqrt{\langle n \rangle}$, such that oscillations are damped out not only at average photon numbers smaller that $\langle n \rangle_{\text{EP}}$ but also in the limit of very large $\langle n \rangle$.

**Nonlinear response for strong perturbations**

We next generalise our study of the regression theorem to the nonlinear regime by successively increasing the reservoir perturbation strength $m_0$ induced by the pulse laser. Figure 4a shows complex eigenvalue spectra $s_\pm(\langle n \rangle)$ obtained from fitting either the linear (top row) or nonlinear (bottom row) expression of $R(t)$ to the recorded condensate time traces. While for the weak perturbation, the linear and nonlinear analysis yield similar results, a deviation from linear response theory is observed for the larger perturbations, as highlighted by the orange shading. In contrast, fitting the nonlinear expression in eq. (1) restores the eigenvalue spectrum of the response dynamics to agree well with the theoretical prediction for the fluctuation dynamics. This improvement demonstrates that the strongly perturbed photon condensate exhibits a pronounced nonlinearity, which occurs in our system when the molecule reservoir is saturated: almost all the excitations that are introduced by the perturbation pulse are then converted into photons leading to a large relative variation of the photon number, that activates the nonlinearity in the stimulated emission and absorption dynamics. Figure 4b shows the residuals $(\sqrt{\sum_i (R_{\text{exp},i} - R_{\text{fit},i})^2/N})/\langle n \rangle$ of the linear and nonlinear fit as the initial pulse perturbation is increased. Here, $N$ denotes the number of recorded samples. Both the linear and the nonlinear fit yield similar and small residuals for weak perturbations below $m_0 \approx 5 \cdot 10^4$ (extracted from the fit), confirming that the condensate-reservoir coupling can here be well described by linear dynamics. Beyond this value, however, the residuals exhibit a splitting. While the linear fit performs significantly worse (i.e., the residuals grow), the nonlinear fit residuals remain close to their initial values, demonstrating that the nonlinear expression is more accurate in describing the response dynamics of the photon condensate in the case of relatively strong perturbations of the molecule reservoir. Despite the improved accuracy of the nonlinear model, we note that for even stronger perturbations, as theoretically expected, also our nonlinear description gradually loses its accuracy due to a linearisation in $m$ (see Methods).

Finally, the nonlinearity of the photon condensate coupled to the molecule reservoir is directly visible when we compare the measured oscillatory response $R(t)$ in Fig. 4c for a weak and a strong perturbation strength with $m_0 \approx 2.7 \cdot 10^4$ and $m_0 \approx 9.3 \cdot 10^4$, respectively. The solid lines show fits based on the linearised and full nonlinear expressions. For the weak perturbation, we find both models to describe the experimental data equally well. For the larger perturbation strength, the measurement is more accurately fitted by the nonlinear expression, as highlighted in the zoom-in view on the 5 to 12 ns time range in the inset of Fig. 4c.

**Discussion**

To conclude, we have measured the response dynamics of a photon Bose-Einstein condensate after a sudden perturbation of the reservoir, which before the perturbation forms an equilibrium steady-state with the condensate. Comparing the response dynamics with the number fluctuations has enabled the experimental verification of the regression theorem for optical quantum gases. Specifically, we have

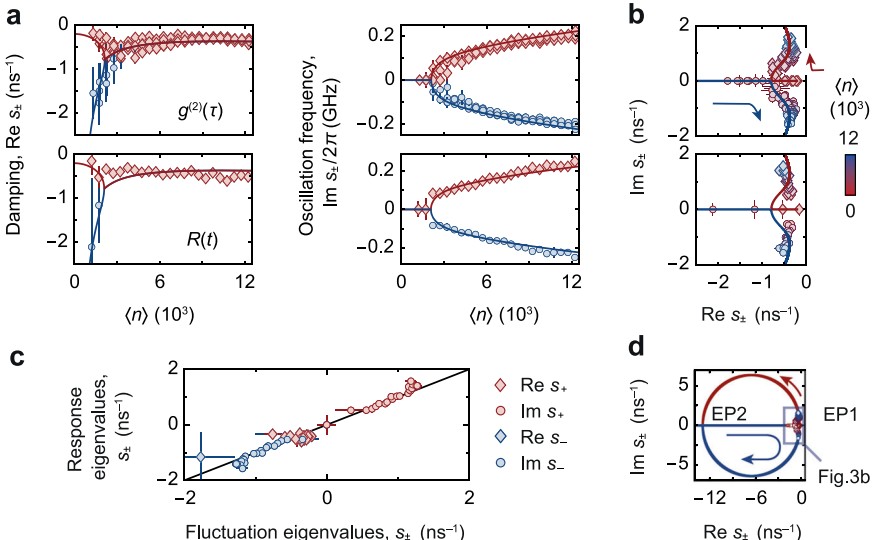

**Fig. 3 | Regression theorem for weak perturbations. a** Damping rate Re($s_\pm$) and oscillation frequency Im($s_\pm$)/($2\pi$) of second-order correlations (top) and perturbation response (bottom) versus $\langle n \rangle$. Red and blue colours indicate $s_+$ and $s_-$, respectively. Solid lines give theory prediction. **b** Complex eigenvalue spectrum near the exceptional point for $g^{(2)}(\tau)$ (left panel) and $R(t)$ (right), along with theory (solid). The eigenvalues evolve as a function of $\langle n \rangle$, as indicated by arrows and symbol colour, from being real to complex-valued. **c** Confirmation of the regression theorem. The fitted eigenvalues $s_\pm$ of the response $R(t)$ are plotted against the fitted eigenvalues of the fluctuations $g^{(2)}(\tau)$ for pairwise corresponding photon numbers $\langle n \rangle$. Symbols and colours indicate four combinations of $s_\pm$, see the legend. Within experimental uncertainties, all points lie on a line of slope one (black line), which shows the agreement of the eigenvalues of response and fluctuations, respectively, and confirms the regression theorem in the optical quantum gas. **d** Imaginary gap opening at exceptional point (EP1) and theoretically predicted gap closing for larger $\langle n \rangle \approx 10^6$ (EP2). Error bars show standard fitting errors.

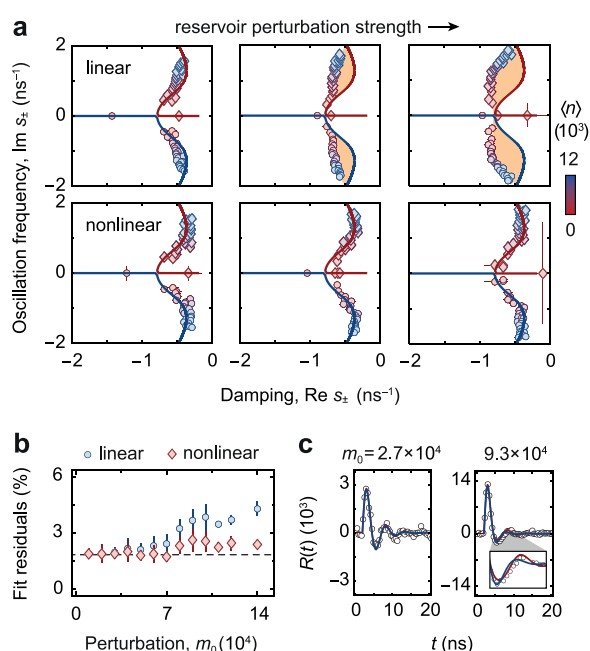

**Fig. 4 | Nonlinear BEC response. a** Complex eigenvalue spectra for increased reservoir perturbations with $\delta n(t_{max})/\langle n \rangle = \{0.59(10), 1.15(23), 1.25(25)\}$ and $m_0 = \{5(2), 9(1), 11(3)\} \cdot 10^4$. Top row: Eigenvalues $s_\pm$ (data points) obtained from a linear fit gradually deviate from theory (solid lines) due to saturation-induced nonlinearity (shaded area). Bottom: $s_\pm$ extracted from the nonlinear fit restores the agreement with theory, demonstrating a beyond-linear regression relation. Diamond symbols correspond to the $s_+$ branch (red line), and circles to the $s_-$ branch (blue line). **b** Residuals of linear and nonlinear fits versus increasing $m_0$ averaged for $\langle n \rangle \geq 4000$ indicate the improved accuracy of the nonlinear fit. The dashed line gives the noise floor. **c** Photon response after a weak (left) and strong perturbation (right) for $\langle n \rangle \approx 8250$, along with linear (blue line) and nonlinear (red) fits. Error bars show standard fitting errors in (**a**), and standard statistical errors in (**b**).

identified a nonlinear response of the BEC at strong driving. Here an extended regression theorem of the form $g^{(2)}(\tau) - 1 \propto d/dt \ln[1 + R(t)/\langle n \rangle]$ holds, which in the limit of small perturbations agrees with the conventional linear form. For the future, the demonstrated scheme to perturb photon condensates constitutes a novel tool for exploring Kibble-Zurek dynamics[40,41], or vortex turbulence[7,36] in optical quantum gases in tailored potentials[42,43]. An extension of the reported single BEC-bath oscillator system to arrays of coherently coupled condensates linked with local reservoirs will enable the exploration of reservoir-induced transport dynamics and open-system topological states[37,38]. Our findings also pave the way for studies of the time-dependent fluctuation-dissipation relation in photon condensates, which so far have only been confirmed for static reactive response functions[29].

## Methods

### Experimental methods and calibrations

For the steady-state excitation of the photon Bose-Einstein condensate inside the dye-filled microcavity, we use a cw laser at 532 nm wavelength (Coherent Verdi 15G). To minimise bleaching of the dye medium (Rhodamine 6G solved in ethylene glycol, concentration 1 mmol L$^{-1}$ and zero-phonon line $\omega_{zpl} = 2\pi \times 550$ THz) as well as pumping to nonradiative triplet states, the pump beam emission is temporally chopped into pulses of 600 ns at a repetition rate of 50 Hz by acousto-optic modulators. During the steady state, a mode-locked laser pulse of 28 ps duration at 532 nm wavelength (EKSPLA PL 2201) is irradiated onto the dye-filled cavity at the same repetition rate, i.e., there is one perturbation pulse during the 600 ns-long steady-state. The laser pulse instantaneously perturbs the number of excited dye molecules, and consequently also drives the photon condensate out of its steady state. The residual condensate emission transmitted through both cavity mirrors is used for the analysis of the experiment. On one side of the cavity, a microscope objective (Mitutoyo MY10X-803) collects the emission to measure the spectral distribution and the spatial intensity distribution of the photon gas. The cavity length is actively stabilised by monitoring the cutoff wavelength behind an Echelle grating on an

EMCCD camera (Andor iXon 897). On the opposite cavity side, the cavity emission is filtered by truncating the high-momentum states of the photon gas using an iris in the far field. The filtered condensate mode is detected using a photomultiplier (PHOTEK PMT 210) with a temporal resolution of 150 ps FWHM that is sampled by an oscilloscope (Tektronix DPO7354C) operated at 20 GSa/s sampling rate with a 2 GHz bandwidth. To calibrate the condensate population $\langle n \rangle$ against the recorded PMT voltage, the photon gas spectrum is fitted with a Bose-Einstein distribution[24].

To exclude systematic sources of errors in our PMT-based measurements of the temporal second-order correlations $g^{(2)}(\tau)$, we perform a benchmark with a HeNe laser at 632.8 nm wavelength, see Supplementary Fig. 1. For the coherent source, one expects $g^{(2)}(\tau) = 1$ for all time delays. However, the obtained signal shows $g^{(2)}(\tau) > 1$ up to $\tau \approx 3$ ns, an observation which we attribute to electronic noise in the detection system. To avoid a misinterpretation of the data arising from this artefact, we exclude data points at $\tau \leq 3$ ns from the analysis of the correlation dynamics. Moreover, radiofrequency noise collected by our detection system (e.g., during pulse picking in the pulse laser system) is visible in the averaged time traces of the photon condensate response at the time of the pulse emission. To mitigate this, an optical delay line has been implemented to shift the pulse arrival time at the microcavity concerning the noise signal. The delay line is realised by a cavity of 24 m length (corresponding to a time delay of 80 ns), which is traversed by the pulse 6 times. To that end, all response measurements can be performed in a temporal region free of residual radiofrequency-induced noise.

## Theoretical model

Using the Kennard-Stepanov relation for the absorption and emission rates of the dye medium, the rate equation for the number of photons $n$ can be rewritten as

$$\frac{dn}{dt} = B_{\text{em}} n \left[ M_\uparrow \left( 1 + \frac{1}{n} + e^{\hbar\Delta/k_{\text{B}}T} \right) - M e^{\hbar\Delta/k_{\text{B}}T} \right] - \kappa n. \quad (2)$$

We represent the number of excited molecules as

$$M_\uparrow = M_{\uparrow,n} + m, \quad (3)$$

where

$$M_{\uparrow,n} = \frac{M + \kappa e^{-\hbar\Delta/k_{\text{B}}T}/B_{\text{em}}}{1 + e^{-\hbar\Delta/k_{\text{B}}T}(1 + 1/\langle n \rangle)} \quad (4)$$

is the number of excited molecules, which corresponds to the steady state with the average photon number equal to $n$. Then for the experimentally relevant case $n \gg 1$, eq. (2) takes the form

$$\frac{dn}{dt} = B_{\text{em}}(1 + e^{\hbar\Delta/k_{\text{B}}T}) mn, \quad (5)$$

which leads to the following nonlinear relation between $m(t)$ and the corresponding evolution of the photon number, starting from its value $\langle n \rangle$ at $t = 0$:

$$n(t) = \langle n \rangle \exp\left[ B_{\text{em}}(1 + e^{\hbar\Delta/k_{\text{B}}T}) \int_0^t m(t')\,dt' \right] \quad (6)$$

To describe the dynamics of $m(t)$ initiated by a sudden perturbation of the excited molecule number at $t = 0$, we utilise the rate

equation for $X \equiv M_{\uparrow,n} + m + n$ with $P = \kappa \langle n \rangle / \langle M_\downarrow \rangle + \Gamma_{\text{sp}} \langle M_\uparrow \rangle / \langle M_\downarrow \rangle$,

$$\frac{dM_{\uparrow,n}}{dt} + \frac{dm}{dt} + \frac{dn}{dt} = -\Gamma_{\text{sp}}\left( M_{\uparrow,n} + m - \langle M_\uparrow \rangle \right) - \kappa(n - \langle n \rangle), \quad (7)$$

where $\langle M_\uparrow \rangle \equiv M_{\uparrow,\langle n \rangle}$ and

$$\frac{dM_{\uparrow,n}}{dt} = \frac{dM_{\uparrow,n}}{dn}\frac{dn}{dt} = \frac{M_{\text{eff}}}{n^2}\frac{dn}{dt} \quad (8)$$

with

$$M_{\text{eff}} = \frac{M + \kappa e^{-\hbar\Delta/k_{\text{B}}T}/B_{\text{em}}}{2[\cosh(\hbar\Delta/k_{\text{B}}T) + 1]}. \quad (9)$$

Inserting eqns. (4), (6) and (8) into eq. (7), one obtains a rather cumbersome nonlinear differential equation for $\int_0^t m\,dt'$, which, in general, cannot be solved analytically. To proceed further, we assume that $m$ is relatively small and hence it can be estimated from the linearised version of the aforementioned equation

$$\frac{dm}{dt} + 2m\delta + \omega_0^2 \int_0^t m\,dt' = 0 \quad (10)$$

with

$$\delta = \frac{\Gamma + \Gamma_{\text{sp}}}{2}, \quad (11)$$

$$\omega_0^2 = \langle n \rangle B_{\text{em}}\left( 1 + e^{\hbar\Delta/k_{\text{B}}T} \right)\left( \kappa + \frac{M_{\text{eff}}\Gamma_{\text{sp}}}{\langle n \rangle^2} \right), \quad (12)$$

where

$$\Gamma = B_{\text{em}}\left( 1 + e^{\hbar\Delta/k_{\text{B}}T} \right)\left( \frac{M_{\text{eff}}}{\langle n \rangle} + \langle n \rangle \right) \quad (13)$$

is the photon number relaxation rate[32].

Equation (10) has the solution

$$\int_0^t m\,dt' = m_0 \frac{e^{s_+ t} - e^{s_- t}}{s_+ - s_-} \quad (14)$$

with

$$s_\pm = -\delta \pm \sqrt{\delta^2 - \omega_0^2}. \quad (15)$$

Here, $m_0 = m(0)$ is the number of molecules excited by the initial perturbation. Inserting eq. (14) into (6), we can express the time dependence of the photon number in an explicit form:

$$n(t) = \langle n \rangle \exp\left[ B_{\text{em}}(1 + e^{\hbar\Delta/k_{\text{B}}T}) m_0 \frac{e^{s_+ t} - e^{s_- t}}{s_+ - s_-} \right] \quad (16)$$

## Relation between $R(t)$ and $g^{(2)}(t)$

The regression theorem implies that the response dynamics $R(t) = n(t) - \langle n \rangle$ of the photon condensate after a perturbation of the molecule reservoir is related to the condensate's second-order coherence $g^{(2)}(t) = \langle n(t)n(0) \rangle / \langle n \rangle^2$ by $g^{(2)}(t) - 1 \propto dR(t)/dt$. In linear response, for small deviations around the mean photon number $n = \langle n \rangle + \Delta n$ and excitation number $X = \langle X \rangle + \Delta X$, we have the following equations of motion for the

time evolution after the system has been perturbed[28]

$$\frac{d}{dt}\begin{pmatrix} \Delta n \\ \Delta X \end{pmatrix} = \begin{pmatrix} -\Gamma & \frac{\langle \delta n^2 \rangle}{M_{\text{eff}}}\Gamma \\ -\kappa + \Gamma_{\text{sp}} & -\Gamma_{\text{sp}} \end{pmatrix}\begin{pmatrix} \Delta n \\ \Delta X \end{pmatrix} \qquad (17)$$

with the steady state photon variance $\langle \delta n^2 \rangle = M_{\text{eff}} \langle n \rangle^2/(M_{\text{eff}} + \langle n \rangle^2)$. The eigenvalues of this matrix correspond to $s_{\pm}$ from eq. (15). Note that we here extend the description of ref. 28 by including losses from the spontaneous decay of the molecules into unconfined modes, which improves the theory description of the experimental observation. The first equation of the matrix form in eq. (17) expresses (i) that deviations in the photon density at constant $X$ relax at the rate $\Gamma$ and (ii) that changes in $X$ lead to a change in the photon number. The second equation of the matrix form expresses that excitations are lost through cavity losses and spontaneous emission.

When an additional laser pulse is applied to perturb the molecules, the system gets initial conditions $\Delta X = \delta X_0$, $\Delta n = 0$ at time $t = 0$. The response to this perturbation is calculated from eq. (17)

$$R(t) = \delta X_0 \frac{\langle \delta n^2 \rangle}{M_{\text{eff}}}\Gamma \frac{e^{s_+ t} - e^{s_- t}}{s_+ - s_-} \qquad (18)$$

and it corresponds to the linearisation of eq. (16) with respect to $m_0$.

To compute the density-density correlator $G^{(2)}(t) = \langle n(t) n(0) \rangle - \langle n \rangle^2 = \langle \delta n(t)\delta n(0)\rangle$ with $\delta n(t) = n(t) - \langle n \rangle$ in the presence of losses, we can use the regression formula:

$$\langle \delta n(t)\delta n(0)\rangle = \langle \delta n(t|\delta n_0)\delta n_0 \rangle \qquad (19)$$

Here $\delta n(t|\delta n_0)$ is the average photon deviation starting from a deviation $\delta n_0$ at time $t = 0$. From eqns. (17) and for $\Delta n = \delta n_0$, $\Delta X = 0$ one finds

$$\delta n(t|\delta n_0) = \delta n_0 \frac{s_+ e^{s_+ t} - s_- e^{s_- t}}{s_+ - s_-}. \qquad (20)$$

From eqns. (19) and (20), we then obtain

$$G^{(2)}(t) = \langle \delta n^2 \rangle \frac{s_+ e^{s_+ t} - s_- e^{s_- t}}{s_+ - s_-}, \qquad (21)$$

where we have set $\delta n_0^2$ to the time-averaged photon variance $\langle \delta n^2 \rangle$. This expression contains the same rates (and frequencies) as the response function in eq. (18) but does not show identical time dependence. From the derivation of the correlation function, one can see that a perturbation in the number of photons at constant $X$ would give a density response with the same time dependence as $G^{(2)}(t)$. Such a perturbation is hard to implement experimentally. By comparing eqns. (18) and (21) one finds

$$G^{(2)}(t) = \frac{M_{\text{eff}}}{\Gamma\,\delta X_0}\frac{d}{dt}R(t). \qquad (22)$$

Using $G^{(2)}(t) = [g^{(2)}(t) - 1]\langle n \rangle^2$, this corresponds to the relation stated at the beginning of this section. In the case of an oscillating density response, the derivative implies that the correlation and response functions have a phase shift of $\pi/2$. Physically, this can be understood from the fact that an external laser pulse excites the molecules, which take some time to be converted into photons. This gives a delay for the response function concerning the correlation function. In Fig. 2 one sees that the correlations show a minimum as a function of time while the response does not. The correlation function $G^{(2)}(t)$ must become negative as can be seen from the differential relation in eq. (22) and $\Delta n(t \rightarrow \infty) = 0$ in the presence of losses. Physically, it is a consequence of a positive photon number fluctuation at some moment to imply larger losses, which reduces the expected photon number later on.

## Linearity of fluctuation dynamics

The particle number fluctuations and the corresponding second-order correlations of a photon Bose-Einstein condensate are governed by linear dynamics even under grand canonical statistical conditions. Here a simple analytical reasoning for this statement is presented. The rate equations for the coupled photon-dye system can be employed to identify two competing rates, which determine whether nonlinear effects in the photon dynamics can or must not be neglected[32]. In the lossless case ($\kappa = \Gamma_{\text{sp}} = 0$), the rate equation for a photon number fluctuation away from its steady-state value reads

$$\frac{d\Delta n}{dt} = - B_{\text{em}}(1 + e^{\hbar\Delta/k_B T})\Delta n^2 - \Gamma(\langle n \rangle)\Delta n, \qquad (23)$$

where $\Gamma(\langle n \rangle)$ is defined in (13) (we indicate here explicitly its photon number dependence). The effective reservoir size $M_{\text{eff}}^0$ is given by eq. (9) with $\kappa = 0$. Nonlinear effects become relevant for $B_{\text{em}}[1 + \exp(\hbar\Delta/k_B T)]\Delta n \sim \Gamma(\langle n \rangle)$. Inserting $\delta n$ (see Section 'Relation of R(t) and $g^{(2)}$(t)'), one has

$$B_{\text{em}}\left(1 + e^{\hbar\Delta/k_B T}\right)\langle n \rangle\sqrt{\frac{M_{\text{eff}}^0}{M_{\text{eff}}^0 + \langle n \rangle^2}} \sim \Gamma(\langle n \rangle). \qquad (24)$$

Supplementary Fig. 2 shows both rates as a function of $\langle n \rangle$ for different dye-cavity detunings, which changes the effective reservoir size. For all curves the second-order correlation rate (r.h.s) exceeds the nonlinear term (l.h.s.), meaning that a photon BEC driven by reservoir-induced fluctuations to good approximation always exhibits linear dynamics. Our experimental data in Figs. 2, 3 confirm this prediction, showing that the second-order correlation dynamics are well described by the derivative of the linear expansion of eq. (1).

## Fitting of experimental data

To analyse the dynamics of the two-time correlations, we fit the second-order correlation data with

$$g^{(2)}(\tau) = a\frac{s_+ e^{s_+(\tau + \Delta\tau)} + s_- e^{s_-(\tau + \Delta\tau)}}{s_+ - s_-} + b, \qquad (25)$$

where the eigenvalues of the dynamics are given by $s_{\pm} = -\delta \pm \sqrt{\delta^2 - \omega_0^2}$. Accordingly, for the response function in the nonlinear form we use the fit function

$$R_{\text{nl}}(t) = a\exp\left[b\frac{e^{s_+(\tau + \Delta\tau)} + e^{s_-(\tau + \Delta\tau)}}{s_+ - s_-}\right] - a \qquad (26)$$

and for the linear form we fit

$$R_{\text{lin}}(t) = a'\frac{e^{s_+(\tau + \Delta\tau)} + e^{s_-(\tau + \Delta\tau)}}{s_+ - s_-}. \qquad (27)$$

The fit parameters $\delta$ and $\omega_0$ resemble the damping constant and natural frequency of a harmonic oscillator; the time delay $\Delta\tau$ together with $a, a', b$ are treated as free parameters for each fit. From the analysis of the condensate response dynamics, we find that both the linear and nonlinear fit can be used to describe the experimental data depending on the strength of the perturbation pulse, as shown in Fig. 4b, c of the main text and in Supplementary Fig. 3.

For a quantitative comparison of the measured eigenvalues $s_{\pm}$ with theory, several system parameters are required: On the one hand, the spontaneous molecule decay rate to unconfined optical modes $\Gamma_{\text{sp}} \approx 200$ MHz and the Einstein coefficient for emission $B_{\text{em}} = 25$ kHz at the dye-cavity detuning $\hbar\Delta/k_B T = -3.87$ (corresponding to a cutoff wavelength $\lambda_c = 570$ nm) are known[28,32]. The molecule number $M$ and

the cavity loss rate $\kappa$, on the other hand, need to be determined. For this, we compare our experimental results to a full numerical solution of the coupled photon-molecule rate equations and minimise the difference by varying the two parameters. To achieve this in a consistent way, all recorded time traces (i.e., for all perturbation powers and all average photon numbers) are fitted simultaneously. This numerical approach is motivated by the fact that even the nonlinear expression for the response dynamics in eq. (1) is not exact, because in its derivation $m$ was assumed to be small. We obtain a molecule number $M = 5.4(15) \cdot 10^9$ and a cavity loss rate $\kappa = 6.4(10)$ GHz. The results are consistent with the corresponding values from analysing the second-order correlation function $g^{(2)}(\tau)$, $M = 6.6(7) \cdot 10^9$ and $\kappa = 6.6(10)$ GHz, which are obtained from fits of the linear expression in eq. (25). As discussed above, this is justified because the fluctuation dynamics are expected to obey linear equations.

Finally, the numerical data allows us to cross-validate the fit results of the linear and nonlinear expressions to the experimental data. Supplementary Fig. 3b shows a map of linear and nonlinear residuals obtained from fitting numerically calculated data as a function of the photon number $\langle n \rangle$ and perturbation strength $m_0$. Fitting the nonlinear expression always yields smaller residuals compared to the linear fit at the respective coordinates; Supplementary Fig. 3c shows the residuals versus $m_0$ when averaged for average photon numbers $\langle n \rangle \geq 4000$, which qualitatively agrees with the experimental observation of Fig. 4b that the nonlinear model can describe the data more accurately at large $m_0$.

## Data availability
The supporting data of this study are available in the Zenodo repository (https://doi.org/10.5281/zenodo.10926250)[44].

## Code availability
Numerical data sets generated during the current study are available from the corresponding author on request.

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

## Acknowledgements

We thank F.E. Öztürk for assistance in early experimental stages; and C. Maes, R. Panico, L. Espert Miranda, N. Longen and A. Redmann for fruitful discussions. This work was financially supported by the DFG within SFB/TR 185 (277625399) and the Cluster of Excellence ML4Q (EXC 2004/1-390534769), and by the DLR with funds provided by the BMWi (50WM1859). J.S. acknowledges support by the EU (ERC, TopoGrand, 101040409), and V.G. and M.Wo. by FWO-Vlaanderen (G061820N). A.E. has received funding from the EU under the Marie Skłodowska-Curie Actions (847471).

## Author contributions

A.S. and G.H. set up the experimental apparatus; A.E. and F.V. contributed to the experimental methods. A.S. performed the measurements and collected the data. A.S. and J.S. analysed the data. V.G. and M.Wo. derived the theoretical model for the description of the photon dynamics. M.We., M.Wo. and J.S. coordinated the project. J.S. wrote the manuscript with contributions from all authors.

## Funding

## Competing interests

The authors declare no competing interests.
