## [Peer Review File · Nature Communications]

Observation of Nonlinear Response and Onsager Regression in a Photon Bose-Einstein CondensateReviewer #1 (Remarks to the Author):

The authors experimentally determine the relation between the fluctuations and the response function of a photon Bose-Einstein condensate. In particular, they show that the two-time photon number correlation function shows the same dynamics and the response of the condensate to an external kick. They thus experimentally demonstrate the quantum regression theorem.

The experimental work is quite impressive and the results are certainly very interesting and important for any researchers studying quantum statistical physics. It therefore, in short, certainly merit publication.

On a technical level, the main text and the methods section contain enough information to understand the details of the experiment. I honestly did not completely understand the theory part of the methods section, but that may be my problem; as far as I can tell, the theory is certainly solid.

There are a few minor points I would suggest the authors clear up.

- Specifically, I do not like the fact that in Fig. 1d, the response dynamics are shown averaged, whereas the fluctuations are shown single shot. The reason I do not like this, is that 1) I cannot see how much fluctuations there are on the dynamics and 2) I cannot see whether there is residual dynamics underneath the single shot. I would therefore advice to include an average and a single shot in both panels.

- Also, I would advice to put a vertical (perhaps dashed) line in the panels in Fig. 2b. The arrow is to far away to judge where the perturbation takes place with respect to the time trace. Obviously, they will align, but it is just easier for the reader if there is a clear line.

- On page four, the phrase "the regime of beyond-linear response" appears. I had to reread that phrase a couple of time, because when I see the words linear and response, I always think linear response. The word beyond simple vanishes. I would therefore expect a phrase like "beyond the regime of linear response".

Reviewer #2 (Remarks to the Author):

The manuscript 'Observation of nonlinear response and Onsager regression in a photon Bose-Einstein condensate' describes the experimental analysis of the reaction of a photon Bose-Einstein condensate to rapid changes in the excitation of the dye molecules environment.

The experimental results are compared with a theoretical analysis, and I do feel that the provided theory is very helpful to understand the experimental observations.

Quantum dynamics in the presence of drive and dissipation is currently of broad interest, and with the results of the manuscript extending beyond equilibrium physics, I am sure that the manuscript will address a broad readership well beyond the field of photon condensation.

I found the physics in the manuscript described well, but I do feel that there are a few instances where the authors might want to clarify some details.

In the last paragraph of page four, there is a relation between the photon number $n(t)$ and the deviation of molecular excitations $m(t)$ from the equilibrium value.

From the discussion in the manuscript, one might be under the impression that this relation was an explicit solution for $n(t)$, but from the discussion in the Supp. Mat. it becomes clear that also $m(t)$ depends on $n(t)$, so that equations of motion for $n(t)$ and for $m(t)$ are coupled.

I appreciate that this dependence can not be discussed in detail here in the manuscript, but a short hint at this interdependence might be helpful for a reader.

On page 5 after Eq.(1), the manuscript refers to two eigenvalues, but it is not specified what is the corresponding operator. It seems that it is the matrix for the classical Hamilton equations of motion of a damped harmonic oscillator, but the manuscript is rather vague about this. I hope that the authors could clarify this (in particular since the eigenvalues are important in the subsequent discussion).

A few lines below, the manuscript refers to the linearization of Eq.(1). Could the authors explain what physical condition justifies linearization?

On page 7, the manuscript states that 'theory hints at a second exceptional point', but it does not clear what theory the authors have in mind. I feel that this would require some clarification.

In Fig 3, most data point match the theory (lines) reasonable well, but in the left top inset showing g^2 , four of the blue data points deviate from theory rather strongly. It would be helpful if the authors could provide some explanation for that.

Overall I enjoyed reading the manuscript, and I would expect that it will attract quite some interest. I am inclined to recommend for publication if the authors can address the above questions in a revised manuscript.

Reviewer #3 (Remarks to the Author):

In this manuscript, Sazhin et al. study experimentally and theoretically the non-equilibrium behavior of a photon condensate. They focus on two observables, the mean response $R(t)$ of the number of photons present in the condensate when pumping is suddenly modified, and the correlation function $g(2)(t)$ at equilibrium.

In the linear response regime for $R(t)$, the authors show that these two quantities have a similar evolution, as predicted by the quantum regression theorem. Their results (figs. 2 and 3) are a direct illustration of this theorem. The authors also study the dynamics of the system in a non-linear response regime and propose an evolution law for $R(t)$ in good agreement with experimental results in this regime.

Overall, this manuscript convincingly shows that this platform is a convenient tool for studying fluctuation-dissipation relations in a complex quantum system. I found it well written, clear and precise. If the authors provide a satisfactory answer to the few points below, I am ready to recommend publication in Nat. Commun.

- page 4, 7 lines before the bottom: the kappa term $\langle n \rangle$ should be divided by $\langle M_{\text{down}} \rangle$.
- The comparison of fits parameters for the data shown in fig. 2ab would be easier for the reader if these data were plotted in a plane (δ , ω), with blue and red colors and corresponding error bars (instead of just giving numerical values in the caption).
- In Figure 3b, the comparison between fits parameters for $g(2)(t)$ and $R(t)$ is difficult. Since it constitutes the main result of the paper, it is important to supplement this figure with another, showing the values of s_+ measured for $g(2)$ as a function of the values of s_+ measured for R (and the same for s_-). Ideally, the points shown in figure 3b should line up on a straight line with slope 1 in this additional figure, and the deviations from this line will be very instructive.

- In figure 4a, the points in the middle column seem to deviate significantly from theory, for both linear and non-linear fits, while the points in the left and right columns are in much better agreement with theory, at least for the non-linear fit. It would be worth mentioning and commenting briefly on this fact.

- On page 7, the authors mention the possible existence of a second exceptional point. Is observation of this point within the scope of the experiment?

The authors experimentally determine the relation between the fluctuations and the response function of a photon Bose-Einstein condensate. In particular, they show that the two-time photon number correlation function shows the same dynamics and the response of the condensate to an external kick. They thus experimentally demonstrate the quantum regression theorem.

The experimental work is quite impressive and the results are certainly very interesting and important for any researchers studying quantum statistical physics. It therefore, in short, certainly merit publication.

On a technical level, the main text and the methods section contain enough information to understand the details of the experiment. I honestly did not completely understand the theory part of the methods section, but that may be my problem; as far as I can tell, the theory is certainly solid.

We would like to thank the reviewer for the positive assessment of our work and the constructive criticism, which helped to clarify the text and the figures.

There are a few minor points I would suggest the authors clear up.

- Specifically, I do not like the fact that in Fig. 1d, the response dynamics are shown averaged, whereas the fluctuations are shown single shot. The reason I do not like this, is that 1) I cannot see how much fluctuations there are on the dynamics and 2) I cannot see whether there is residual dynamics underneath the single shot. I would therefore advice to include an average and a single shot in both panels.

As suggested by the reviewer, we have added two more panels to Fig. 1d, which now shows the average and the single shot time traces for both time windows. First, the top row shows the time window around 230 ns at which the perturbation is irradiated to the bath; the window is used to analyse the mean response $R(t)$. Despite the fluctuations, one can see here an indication that the perturbation leads to a somewhat increased photon number just before $t = 230$ ns. A more clearly visible response signal, however, only becomes evident after averaging hundreds of time traces. Secondly, the bottom row now gives both single shot and averaged data in the window at later times; this window is used for the analysis of the second-order correlation function $g^{(2)}(\tau)$. Here, the average signal is essentially constant without any temporal structure.

- Also, I would advice to put a vertical (perhaps dashed) line in the panels in Fig. 2b. The arrow is too far away to judge where the perturbation takes place with respect to the time trace. Obviously, they will align, but it is just easier for the reader if there is a clear line.

We have added a dashed line to both panels in Fig. 2b to indicate the time when the pulse perturbs the reservoir and modified the figure caption accordingly.

- On page four, the phrase "the regime of beyond-linear response" appears. I had to reread that phrase a couple of time, because when I see the words linear and response, I always think linear response. The word beyond simple vanishes. I would therefore expect a phrase like "beyond the regime of linear response".

We have modified the expression, which has indeed helped to remove this ambiguity.

The manuscript 'Observation of nonlinear response and Onsager regression in a photon Bose-Einstein condensate' describes the experimental analysis of the reaction of a photon Bose-Einstein condensate to rapid changes in the excitation of the dye molecules environment. The experimental results are compared with a theoretical analysis, and I do feel that the provided theory is very helpful to understand the experimental observations.

Quantum dynamics in the presence of drive and dissipation is currently of broad interest, and with the results of the manuscript extending beyond equilibrium physics, I am sure that the manuscript will address a broad readership well beyond the field of photon condensation.

I found the physics in the manuscript described well, but I do feel that there are a few instances where the authors might want to clarify some details.

We thank the reviewer for the constructive criticism and appreciate the positive assessment of our work. In the following we respond to the specific remarks raised by the reviewer.

In the last paragraph of page four, there is a relation between the photon number $n(t)$ and the deviation of molecular excitations $m(t)$ from the equilibrium value. From the discussion in the manuscript, one might be under the impression that this relation was an explicit solution for $n(t)$, but from the discussion in the Supp. Mat. it becomes clear that also $m(t)$ depends on $n(t)$, so that equations of motion for $n(t)$ and for $m(t)$ are coupled. I appreciate that this dependence can not be discussed in detail here in the manuscript, but a short hint at this interdependence might be helpful for a reader.

We agree that this point deserved some clarification. We have added a sentence commenting on the interplay between the photon and excitation number dynamics.

On page 5 after Eq. (1), the manuscript refers to two eigenvalues, but it is not specified what is the corresponding operator. It seems that it is the matrix for the classical Hamilton equations of motion of a damped harmonic oscillator, but the manuscript is rather vague about this. I hope that the authors could clarify this (in particular since the eigenvalues are important in the subsequent discussion).

The reviewer is perfectly right that the eigenvalues were introduced somewhat abruptly without a proper definition of the corresponding matrix. We refer to the eigenvalues of the matrix that describes the coupled mean-field equations of motion for the variation of the photon and excited molecule number, which we have added to the Methods following the reviewer's remark. The eigenvalues $s_{\pm} = -\delta \pm \sqrt{\delta^2 - \omega_0^2}$ of this non-Hermitian matrix are in general complex-valued numbers, which coalesce at $\delta = \omega_0$, meaning that $s_+ = s_-$. The physical origin of the exceptional point is the openness of the system (here, specifically, caused by the cavity losses). Due to the openness it is important to note that one generally expects such exceptional points also in the energy levels. As we do not perform spectroscopy but only focus on the mean-field dynamics, such a study lies beyond the scope of the present work, but it remains an important and interesting topic for future work.

We have added a sentence to clarify the origin of the eigenvalues in the main text and now explicitly give the matrix in eq. (S16) in the Methods.

A few lines below, the manuscript refers to the linearization of Eq. (1). Could the authors explain what physical condition justifies linearization?

The linearisation is applicable for weak perturbations m_0 at time $t = 0$. By inspecting eq. (1), one can see that the exponent is determined by m_0 and by the system parameters B_{em} , Δ , T , s_{\pm} which are all

constant for a fixed $\langle n \rangle$. Thus, the value of the exponent is ultimately only controlled by m_0 , justifying that the linearisation of eq. (1) is valid for small perturbations. Experimentally, this is precisely what we observe in Fig. 4a, where for the weakest perturbation (leftmost panels) the linear and nonlinear expression yield similar eigenvalues s_{\pm} and both agree with the eigenvalue theory (solid line). Only as the perturbation strength is increased, the expressions yield different results. Figure 4b shows essentially the same behaviour, but here based on the fit residuals of the linear and nonlinear expression fit, which gradually separate as m_0 is increased. This confirms that the linearisation is justified for weak perturbations. Finally, we note that the response dynamics smoothly crosses over from linear to nonlinear behaviour, *i.e.*, there is no sharp transition point at a specific m_0 below which the exponential term in eq. (1) can be expanded for an adequate description of the dynamics.

We have added a half-sentence in the paragraph following eq. (1), stating that the linearisation is valid for weak perturbations.

On page 7, the manuscript states that 'theory hints at a second exceptional point', but it does not clear what theory the authors have in mind. I feel that this would require some clarification.

We agree with the reviewer that the referral to 'theory' was not precise enough. We refer to our own theory model discussed in the Methods, specifically eqns. (S10)-(S12). The equations allow one to calculate the exceptional points based on the condition $\delta(\langle n \rangle) = \omega_0(\langle n \rangle)$ (or equivalently $s_+(\langle n \rangle) = s_-(\langle n \rangle)$). Due to the different functional form of $\delta(\langle n \rangle)$ and $\omega_0(\langle n \rangle)$, both functions intersect twice: The 'first' exceptional point occurs at average photon numbers $\langle n \rangle \approx 2000$. Here the system transitions from biexponentially ($s_{\pm} \in \mathbb{R}$) to oscillatory damped dynamics ($s_{\pm} \in \mathbb{C}$). The 'second' exceptional point occurs at $\langle n \rangle \approx 10^6$. Here the condensate system re-enters the biexponentially damped state ($s_{\pm} \in \mathbb{R}$).

We have revised the discussion of Fig. 3d which shows both exceptional points. This should clarify the theoretical origin of the exceptional points.

In Fig 3, most data point match the theory (lines) reasonable well, but in the left top inset showing $g^{(2)}$, four of the blue data points deviate from theory rather strongly. It would be helpful if the authors could provide some explanation for that.

The reviewer correctly points out that a few data points obtained from the $g^{(2)}(\tau)$ measurement were deviating from the theory prediction for small condensate numbers $\langle n \rangle$. In the revised version of the manuscript, we have modified Fig. 3, following the suggestion by Reviewer 3 to add a plot showing the correlation between the experimentally determined eigenvalues s_{\pm} for the fluctuation and the independently determined eigenvalues s_{\pm} for the response, thereby providing a conclusive (and visually more appealing) confirmation of the regression theorem.

Along with these substantial revisions, we have re-evaluated all datasets and performed a careful error analysis. In the revision process, the deviations mentioned by Reviewer 2 disappeared. Still, it is evident from the error bars that the experimental uncertainties are much larger at small $\langle n \rangle$. This can be understood from the longer recording times required for averaging the low-signal data throughout which experimental imperfections, *e.g.*, cavity drifts or radiofrequency noise, can affect the data quality. We have taken considerable measures to mitigate radiofrequency-induced noise artefacts (see Methods), but in the low-photon-number regime our measurement accuracy is simply limited by such systematics. Despite the uncertainties and after careful re-analysis of the data, we believe that our measurements over a wide range of photon numbers provide a solid confirmation of the regression theorem in the optical quantum gas.

Overall I enjoyed reading the manuscript, and I would expect that it will attract quite some interest. I am inclined to recommend for publication if the authors can address the above questions in a revised manuscript.

We are pleased about the appreciation and thank the reviewer once more for the insightful comments on

our manuscript.

In this manuscript, Sazhin et al. study experimentally and theoretically the non-equilibrium behavior of a photon condensate. They focus on two observables, the mean response $R(t)$ of the number of photons present in the condensate when pumping is suddenly modified, and the correlation function $g^{(2)}(t)$ at equilibrium.

In the linear response regime for $R(t)$, the authors show that these two quantities have a similar evolution, as predicted by the quantum regression theorem. Their results (figs. 2 and 3) are a direct illustration of this theorem. The authors also study the dynamics of the system in a non-linear response regime and propose an evolution law for $R(t)$ in good agreement with experimental results in this regime.

Overall, this manuscript convincingly shows that this platform is a convenient tool for studying fluctuation-dissipation relations in a complex quantum system. I found it well written, clear and precise. If the authors provide a satisfactory answer to the few points below, I am ready to recommend publication in Nat. Commun.

We thank the reviewer for the constructive criticism and various suggestions, which helped to improve the manuscript. In the following we respond to the specific remarks raised by the reviewer.

- page 4, 7 lines before the bottom: the kappa term $\langle n \rangle$ should be divided by $\langle M_{\text{down}} \rangle$.

We thank the reviewer for pointing us at this mistake. We have corrected it.

- The comparison of fits parameters for the data shown in Fig. 2ab would be easier for the reader if these data were plotted in a plane (δ, ω) , with blue and red colors and corresponding error bars (instead of just giving numerical values in the caption).

We agree with the reviewer that giving only numbers for the fit parameters was not very intuitive. Following the reviewer's suggestion, we have added Fig. 2c, which shows the fit parameters in the δ - ω_0 plane and visualises their agreement for both photon numbers. Moreover, the representation now highlights the conditions for oscillatory ($\delta < \omega_0$) and biexponential relaxation dynamics ($\delta > \omega_0$).

- In Figure 3b, the comparison between fits parameters for $g^{(2)}(t)$ and $R(t)$ is difficult. Since it constitutes the main result of the paper, it is important to supplement this figure with another, showing the values of s_+ measured for $g^{(2)}$ as a function of the values of s_+ measured for R (and the same for s_-). Ideally, the points shown in figure 3b should line up on a straight line with slope 1 in this additional figure, and the deviations from this line will be very instructive.

We thank the reviewer for the excellent suggestion, which substantially improved the manuscript. After carefully revising the data from Fig. 3, we now give an additional plot in Fig. 3c which directly relates the fitted eigenvalues s_{\pm} from the response (y-axis) to the s_{\pm} values from the fluctuations (x-axis). For clarity, we are comparing four sets of data: $\text{Re } s_+$ (red diamonds), $\text{Im } s_+$ (red circles), $\text{Re } s_-$ (blue diamonds), and $\text{Im } s_-$ (blue circles). One can see how the data points line up on top of a straight line of slope 1. As suggested by the reviewer, such a representation is instructive and strengthens the main point of the study, which is the confirmation of the regression theorem.

We have revised Fig. 3 and expanded on its discussion in the main text.

- In figure 4a, the points in the middle column seem to deviate significantly from theory, for both linear and non-linear fits, while the points in the left and right columns are in much better agreement with theory, at least for the non-linear fit. It would be worth mentioning and commenting briefly on this fact.

We thank the reviewer for hinting at this discrepancy. We have revised the plots from Fig. 4a, and realised that there was a problem in the theory curves which were missing the spontaneous decay term, see eqns. (S10) and (S11) of the Methods. Including the term has slightly shifted the theory curves with respect to the data, which now agree very well with each other. All theory curves for s_{\pm} shown in the manuscript are now based on these equations and use the same parameters which are given in the Methods.

Finally, we would like to comment on the seeming deviation of the nonlinear data from theory in the vicinity of the exceptional point in Fig. 4a (central bottom panel, intermediate perturbation strength), as also noted by the reviewer. The real-valued data points actually perfectly agree with theory; to appreciate this, note that the diamond symbols indicate s_+ eigenvalues (red branch), and the circles indicate the s_- eigenvalues (blue branch). Perhaps this was not clear. We have added a comment on the symbols to clarify this point.

- On page 7, the authors mention the possible existence of a second exceptional point. Is observation of this point within the scope of the experiment?

The observation of the second exceptional point is certainly an interesting topic for future studies. For the presently investigated system parameters its observation is not within the scope of the experiment, because the average condensate number at which the second exceptional point is expected ($\langle n \rangle_{\text{EP},2} \approx 10^6$ photons) cannot be realised in a controlled way because the photon gas exhibits a macroscopic occupation of several energetically higher-lying trapped levels in the limit of such large photon numbers. In other words, the system goes multi-mode. For the future, it may however become possible to optimise the system parameters (in particular, by increasing the molecule number M , and reducing both the dye-cavity detuning Δ and the cavity losses κ) so as to reduce $\langle n \rangle_{\text{EP},2}$ to a regime of accessible condensate populations; for example, for $M = 2 \cdot 10^9$, $\Delta = 0$, $\kappa = 1$ GHz, one finds $\langle n \rangle_{\text{EP},1} \approx 25\,000$ and $\langle n \rangle_{\text{EP},2} \approx 50\,000$.

Reviewer #1 (Remarks to the Author):

In my opinion, the authors have responded adequately to my comments and those of the other reviewers and have made the appropriate changes to the manuscript. I can therefore be very brief and recommend the manuscript for publication!

Reviewer #2 (Remarks to the Author):

The authors have addressed the questions raised in my previous report, and I feel that the modifications to the manuscript are very helpful for clarification. I am happy to recommend for publication of the manuscript.

Reviewer #3 (Remarks to the Author):

In this revised version, the authors have adequately addressed all points that I raised in their initial submission. I therefore recommend that this paper be published in its present form in Nature Communications.